# Interlaced Greedy Algorithm for Maximization of Submodular Functions in Nearly Linear Time

**Alan Kuhnle**
Department of Computer Science
Florida State University
Tallahassee, FL 32306
akuhnle@fsu.edu

## Abstract

A deterministic approximation algorithm is presented for the maximization of non-monotone submodular functions over a ground set of size $n$ subject to cardinality constraint $k$; the algorithm is based upon the idea of interlacing two greedy procedures. The algorithm uses interlaced, thresholded greedy procedures to obtain tight ratio $1/4 - \varepsilon$ in $O\left(\frac{n}{\varepsilon} \log\left(\frac{k}{\varepsilon}\right)\right)$ queries of the objective function, which improves upon both the ratio and the quadratic time complexity of the previously fastest deterministic algorithm for this problem. The algorithm is validated in the context of two applications of non-monotone submodular maximization, on which it outperforms the fastest deterministic and randomized algorithms in prior literature.

## 1 Introduction

A nonnegative function $f$ defined on subsets of a ground set $U$ of size $n$ is *submodular* iff for all $A, B \subseteq U$, $x \in U \setminus B$, such that $A \subseteq B$, it holds that $f(B \cup x) - f(B) \leq f(A \cup x) - f(A)$. Intuitively, the property of submodularity captures diminishing returns. Because of a rich variety of applications, the maximization of a nonnegative submodular function with respect to a cardinality constraint (MCC) has a long history of study (Nemhauser et al., 1978). Applications of MCC include viral marketing (Kempe et al., 2003), network monitoring (Leskovec et al., 2007), video summarization (Mirzasoleiman et al., 2018), and MAP Inference for Determinantal Point Processes (Gillenwater et al., 2012), among many others. In recent times, the amount of data generated by many applications has been increasing exponentially; therefore, linear or sublinear-time algorithms are needed.

If a submodular function $f$ is monotone[1], greedy approaches for MCC have proven effective and nearly optimal, both in terms of query complexity and approximation factor: subject to a cardinality constraint $k$, a simple greedy algorithm gives a $(1 - 1/e)$ approximation ratio in $O(kn)$ queries (Nemhauser et al., 1978), where $n$ is the size of the instance. Furthermore, this ratio is optimal under the value oracle model (Nemhauser and Wolsey, 1978). Badanidiyuru and Vondrák (2014) sped up the greedy algorithm to require $O\left(\frac{n}{\varepsilon} \log \frac{n}{\varepsilon}\right)$ queries while sacrificing only a small $\varepsilon > 0$ in the approximation ratio, while Mirzasoleiman et al. (2015) developed a randomized $(1 - 1/e - \varepsilon)$ approximation in $O(n/\varepsilon)$ queries.

When $f$ is non-monotone, the situation is very different; no subquadratic deterministic algorithm has yet been developed. Although a linear-time, randomized $(1/e - \varepsilon)$-approximation has been developed by Buchbinder et al. (2015), which requires $O\left(\frac{n}{\varepsilon^2} \log \frac{1}{\varepsilon}\right)$ queries, the performance guarantee of this algorithm holds only in expectation. A derandomized version of the algorithm with ratio $1/e$ has been

Table 1: Fastest algorithms for cardinality constraint

| Algorithm | Ratio | Time complexity | Deterministic? |
|---|---|---|---|
| FastInterlaceGreedy (Alg. 2) | $1/4 - \varepsilon$ | $O\left(\frac{n}{\varepsilon} \log \frac{k}{\varepsilon}\right)$ | Yes |
| Gupta et al. (2010) | $1/6 - \varepsilon$ | $O\left(nk + \frac{n}{\varepsilon}\right)$ | Yes |
| Buchbinder et al. (2015) | $1/e - \varepsilon$ | $O\left(\frac{n}{\varepsilon^2} \log \frac{1}{\varepsilon}\right)$ | No |

developed by Buchbinder and Feldman (2018a) but has time complexity $O(k^3 n)$. Therefore, in this work, an emphasis is placed upon the development of nearly linear-time, deterministic approximation algorithms.

**Contributions**

The deterministic approximation algorithm InterlaceGreedy (Alg. 1) is provided for maximization of a submodular function subject to a cardinality constraint (MCC). InterlaceGreedy achieves ratio $1/4$ in $O(kn)$ queries to the objective function. A faster version of the algorithm is formulated in FastInterlaceGreedy (Alg. 2), which achieves ratio $(1/4 - \varepsilon)$ in $O\left(\frac{n}{\varepsilon} \log \frac{k}{\varepsilon}\right)$ queries. In Table 1, the relationship is shown to the fastest deterministic and randomized algorithms for MCC in prior literature.

Both algorithms operate by interlacing two greedy procedures together in a novel manner; that is, the two greedy procedures alternately select elements into disjoint sets and are disallowed from selection of the same element. This technique is demonstrated first with the interlacing of two standard greedy procedures in InterlaceGreedy, before interlacing thresholded greedy procedures developed by Badanidiyuru and Vondrák (2014) for monotone submodular functions to obtain the algorithm FastInterlaceGreedy.

The algorithms are validated in the context of cardinality-constrained maximum cut and social network monitoring, which are both instances of MCC. In this evaluation, FastInterlaceGreedy is more than an order of magnitude faster than the fastest deterministic algorithm (Gupta et al., 2010) and is both faster and obtains better solution quality than the fastest randomized algorithm (Buchbinder et al., 2015). The source code for all implementations is available at `https://gitlab.com/kuhnle/non-monotone-max-cardinality`.

**Organization**    The rest of this paper is organized as follows. Related work and preliminaries on submodular optimization are discussed in the rest of this section. In Section 2, InterlaceGreedy and FastInterlaceGreedy are presented and analyzed. Experimental validation is provided in Section 4.

**Related Work**

The literature on submodular optimization comprises many works. In this section, a short review of relevant techniques is given for MCC; that is, maximization of non-monotone, submodular functions over a ground set of size $n$ with cardinality constraint $k$. For further information on other types of submodular optimization, interested readers are directed to the survey of Buchbinder and Feldman (2018b) and references therein.

A deterministic local search algorithm was developed by Lee et al. (2010), which achieves ratio $1/4 - \varepsilon$ in $O(n^4 \log n)$ queries. This algorithm runs two approximate local search procedures in succession. By contrast, the algorithm FastInterlaceGreedy employs interlacing of greedy procedures to obtain the same ratio in $O\left(\frac{n}{\varepsilon} \log \frac{k}{\varepsilon}\right)$ queries. In addition, a randomized local search algorithm was formulated by Vondrák (2013), which achieves ratio $\approx 0.309$ in expectation.

Gupta et al. (2010) developed a deterministic, iterated greedy approach, wherein two greedy procedures are run in succession and an algorithm for unconstrained submodular maximization are employed. This approach requires $O(nk)$ queries and has ratio $1/(4 + \alpha)$, where $\alpha$ is the inverse ratio of the employed subroutine for unconstrained, non-monotone submodular maximization; under the value query model, the smallest possible value for $\alpha$ is 2, as shown by Feige et al. (2011), so this ratio is at most $1/6$. The iterated greedy approach of Gupta et al. (2010) first runs one standard greedy algorithm to completion, then starts a second standard greedy procedure; this differs from the interlacing procedure which runs two greedy procedures concurrently and alternates between

the selection of elements. The algorithm of Gupta et al. (2010) is experimentally compared to FastInterlaceGreedy in Section 4. The iterated greedy approach of Gupta et al. (2010) was extended and analyzed under more general constraints by a series of works: Mirzasoleiman et al. (2016); Feldman et al. (2017); Mirzasoleiman et al. (2018).

An elegant randomized greedy algorithm of Buchbinder et al. (2014) achieves expected ratio $1/e$ in $O(kn)$ queries for MCC; this algorithm was derandomized by Buchbinder and Feldman (2018a), but the derandomized version requires $O\left(k^3 n\right)$ queries. The randomized version was sped up in Buchbinder et al. (2015) to achieve expected ratio $1/e - \varepsilon$ and require $O\left(\frac{n}{\varepsilon^2} \log \frac{1}{\varepsilon}\right)$ queries. Although this algorithm has better time complexity than FastInterlaceGreedy, the ratio of $1/e - \varepsilon$ holds only in expectation, which is much weaker than a deterministic approximation ratio. The algorithm of Buchbinder et al. (2015) is experimentally evaluated in Section 4.

Recently, an improvement in the adaptive complexity of MCC was made by Balkanski et al. (2018). Their algorithm, BLITS, requires $O\left(\log^2 n\right)$ adaptive rounds of queries to the objective, where the queries within each round are independent of one another and thus can be parallelized easily. Previously the best adaptivity was the trivial $O(n)$. However, each round requires $\Omega(OPT^2)$ samples to approximate expectations, which for the applications evaluated in Section 4 is $\Omega(n^4)$. For this reason, BLITS is evaluated as a heuristic in comparison with the proposed algorithms in Section 4. Further improvements in adaptive complexity have been made by Fahrbach et al. (2019) and Ene and Nguyen (2019).

Streaming algorithms for MCC make only one or a few passes through the ground set. Streaming algorithms for MCC include those of Chekuri et al. (2015); Feldman et al. (2018); Mirzasoleiman et al. (2018). A streaming algorithm with low adaptive complexity has recently been developed by Kazemi et al. (2019). In the following, the algorithms are allowed to make an arbitrary number of passes through the data.

Currently, the best approximation ratio of any algorithm for MCC is $0.385$ of Buchbinder and Feldman (2016). Their algorithm also works under a more general constraint than cardinality constraint; namely, a matroid constraint. This algorithm is the latest in a series of works (e.g. (Naor and Schwartz, 2011; Ene and Nguyen, 2016)) using the multilinear extension of a submodular function, which is expensive to evaluate.

**Preliminaries**

Given $n \in \mathbb{N}$, the notation $[n]$ is used for the set $\{0, 1, \ldots, n - 1\}$. In this work, functions $f$ with domain all subsets of a finite set are considered; hence, without loss of generality, the domain of the function $f$ is taken to be $2^{[n]}$, which is all subsets of $[n]$. An equivalent characterization of submodularity is that for each $A, B \subseteq [n]$, $f(A \cup B) + f(A \cap B) \leq f(A) + f(B)$. For brevity, the notation $f_x(A)$ is used to denote the marginal gain $f(A \cup \{x\}) - f(A)$ of adding element $x$ to set $A$.

In the following, the problem studied is to maximize a submodular function under a cardinality constraint (MCC), which is formally defined as follows. Let $f : 2^n \to \mathbb{R}^+$ be submodular; let $k \in [n]$. Then the problem is to determine

$$\underset{A \subseteq [n] : |A| \leq k}{\arg \max} \ f(A).$$

An instance of MCC is the pair $(f, k)$; however, rather than an explicit description of $f$, the function $f$ is accessed by a value oracle; the value oracle may be queried on any set $A \subseteq [n]$ to yield $f(A)$. The efficiency or runtime of an algorithm is measured by the number of queries made to the oracle for $f$.

Finally, without loss of generality, instances of MCC considered in the following satisfy $n \geq 4k$. If this condition does not hold, the function may be extended to $[m]$ by adding dummy elements to the domain which do not change the function value. That is, the function $g : 2^m \to \mathbb{R}^+$ is defined as $g(A) = f(A \cap [n])$; it may be easily checked that $g$ remains submodular, and any possible solution to the MCC instance $(g, k)$ maps[2] to a solution of $(f, k)$ of the same value. Hence, the ratio of any solution to $(g, k)$ to the optimal is the same as the ratio of the mapped solution to the optimal on $(f, k)$.

# 2 Approximation Algorithms

In this section, the approximation algorithms based upon interlacing greedy procedures are presented. In Section 2.1, the technique is demonstrated with standard greedy procedures in algorithm InterlaceGreedy. In Section 2.2, the nearly linear-time algorithm FastInterlaceGreedy is introduced.

## 2.1 The InterlaceGreedy Algorithm

In this section, the InterlaceGreedy algorithm (InterlaceGreedy, Alg. 1) is introduced. InterlaceGreedy takes as input an instance of MCC and outputs a set $C$.

---

**Algorithm 1** InterlaceGreedy $(f, k)$: The InterlaceGreedy Algorithm

1: **Input:** $f : 2^{[n]} \to \mathbb{R}^+$, $k \in [n]$
2: **Output:** $C \subseteq [n]$, such that $|C| \leq k$.
3: $A_0 \leftarrow B_0 \leftarrow \emptyset$
4: **for** $i \leftarrow 0$ to $k-1$ **do**
5:      $a_i \leftarrow \arg\max_{x \in [n] \setminus (A_i \cup B_i)} f_x(A_i)$
6:      $A_{i+1} \leftarrow A_i + a_i$
7:      $b_i \leftarrow \arg\max_{x \in [n] \setminus (A_{i+1} \cup B_i)} f_x(B_i)$
8:      $B_{i+1} \leftarrow B_i + b_i$
9: $D_1 \leftarrow E_1 \leftarrow \{a_0\}$
10: **for** $i \leftarrow 1$ to $k-1$ **do**
11:      $d_i \leftarrow \arg\max_{x \in [n] \setminus (D_i \cup E_i)} f_x(D_i)$
12:      $D_{i+1} \leftarrow D_i + d_i$
13:      $e_i \leftarrow \arg\max_{x \in [n] \setminus (D_{i+1} \cup E_i)} f_x(E_i)$
14:      $E_{i+1} \leftarrow E_i + e_i$
15: **return** $C \leftarrow \arg\max\{f(A_i), f(B_i), f(D_i), f(E_i) : i \in [k+1]\}$

---

InterlaceGreedy operates by interlacing two standard greedy procedures. This interlacing is accomplished by maintaining two disjoint sets $A$ and $B$, which are initially empty. For $k$ iterations, the element $a \notin B$ with the highest marginal gain with respect to $A$ is added to $A$, followed by an analogous greedy selection for $B$; that is, the element $b \notin A$ with the highest marginal gain with respect to $B$ is added to $B$. After the first set of interlaced greedy procedures complete, a modified version is repeated with sets $D, E$, which are initialized to the maximum-value singleton $\{a_0\}$. Finally, the algorithm returns the set with the maximum $f$-value of any query the algorithm has made to $f$.

If $f$ is submodular, InterlaceGreedy has an approximation ratio of $1/4$ and query complexity $O(kn)$; the deterministic algorithm of Gupta et al. (2010) has the same time complexity to achieve ratio $1/6$. The full proof of Theorem 1 is provided in Appendix A.

**Theorem 1.** *Let $f : 2^{[n]} \to \mathbb{R}^+$ be submodular, let $k \in [n]$, let $O = \arg\max_{|S| \leq k} f(S)$, and let $C = InterlaceGreedy (f, k)$. Then*

$$f(C) \geq f(O)/4,$$

*and InterlaceGreedy makes $O(kn)$ queries to $f$.*

*Proof sketch.* The argument of Fisher et al. (1978) shows that the greedy algorithm is a $(1/2)$-approximation for monotone submodular maximization with respect to a matroid constraint. This argument also applies to non-monotone, submodular functions, but it shows only that $f(S) \geq \frac{1}{2}f(O \cup S)$, where $S$ is returned by the greedy algorithm. Since $f$ is non-monotone, it is possible for $f(O \cup S) < f(S)$. The main idea of the InterlaceGreedy algorithm is to exploit the fact that if $S$ and $T$ are disjoint,

$$f(O \cup S) + f(O \cup T) \geq f(O) + f(O \cup S \cup T) \geq f(O), \tag{1}$$

which is a consequence of the submodularity of $f$. Therefore, by interlacing two greedy procedures, two disjoint sets $A, B$ are obtained, which can be shown to almost satisfy $f(A) \geq \frac{1}{2}f(O \cup A)$ and $f(B) \geq \frac{1}{2}f(O \cup B)$, after which the result follows from (1). There is a technicality wherein the element $a_0$ must be handled separately, which requires the second round of interlacing to address. □

## 2.2 The FastInterlaceGreedy Algorithm

In this section, a faster interlaced greedy algorithm (FastInterlaceGreedy (FIG), Alg. 2) is formulated, which requires $O(n \log k)$ queries. As input, an instance $(f, k)$ of MCC is taken, as well as a parameter $\delta > 0$.

---

**Algorithm 2** FIG $(f, k, \delta)$: The FastInterlaceGreedy Algorithm

---

1: **Input:** $f : 2^{[n]} \to \mathbb{R}^+$, $k \in [n]$
2: **Output:** $C \subseteq [n]$, such that $|C| \leq k$.
3: $A_0 \leftarrow B_0 \leftarrow \emptyset$
4: $M \leftarrow \tau_A \leftarrow \tau_B \leftarrow \max_{x \in [n]} f(x)$
5: $i \leftarrow -1, a_{-1} \leftarrow 0, b_{-1} \leftarrow 0$
6: **while** $\tau_A \geq \delta M/k$ or $\tau_B \geq \delta M/k$ **do**
7: $\quad (a_{i+1}, \tau_A) \leftarrow \text{ADD}(A, B, a_i, \tau_A)$
8: $\quad (b_{i+1}, \tau_B) \leftarrow \text{ADD}(B, A, b_i, \tau_B)$
9: $\quad i \leftarrow i + 1$
10: $D_1 \leftarrow E_1 \leftarrow \{a_0\}, \tau_D \leftarrow \tau_E \leftarrow M$
11: $i \leftarrow 0, d_0 \leftarrow 0, e_0 \leftarrow 0$
12: **while** $\tau_D \geq \delta M/k$ or $\tau_E \geq \delta M/k$ **do**
13: $\quad (d_{i+1}, \tau_D) \leftarrow \text{ADD}(D, E, d_i, \tau_D)$
14: $\quad (e_{i+1}, \tau_E) \leftarrow \text{ADD}(E, D, e_i, \tau_E)$
15: $\quad i \leftarrow i + 1$
16: **return** $C \leftarrow \arg \max \{f(A), f(B), f(D), f(E)\}$

---

**Algorithm 3** ADD $(S, T, j, \tau)$: The ADD subroutine

---

1: **Input:** Two sets $S, T \subseteq [n]$, element $j \in [n]$, $\tau \in \mathbb{R}^+$
2: **Output:** $(i, \tau)$, such that $i \in [n]$, $\tau \in \mathbb{R}^+$
3: **if** $|S| = k$ **then**
4: $\quad$ **return** $(0, (1 - \delta)\tau)$
5: **while** $\tau \geq \delta M/k$ **do**
6: $\quad$ **for** $(x \leftarrow j; x < n; x \leftarrow x + 1)$ **do**
7: $\quad\quad$ **if** $x \notin T$ **then**
8: $\quad\quad\quad$ **if** $f_x(S) \geq \tau$ **then**
9: $\quad\quad\quad\quad S \leftarrow S \cup \{x\}$
10: $\quad\quad\quad\quad$ **return** $(x, \tau)$
11: $\quad \tau \leftarrow (1 - \delta)\tau$
12: $\quad j \leftarrow 0$
13: **return** $(0, \tau)$

---

The algorithm FIG works as follows. As in InterlaceGreedy, there is a repeated interlacing of two greedy procedures. However, to ensure a faster query complexity, these greedy procedures are thresholded: a separate threshold $\tau$ is maintained for each of the greedy procedures. The interlacing is accomplished by alternating calls to the ADD subroutine (Alg. 3), which adds a single element and is described below. When all of the thresholds fall below the value $\delta M/k$, the maximum of the greedy solutions is returned; here, $\delta > 0$ is the input parameter, $M$ is the maximum value of a singleton, and $k \leq n$ is the cardinality constraint.

The ADD subroutine is responsible for adding a single element above the input threshold and decreasing the threshold. It takes as input four parameters: two sets $S, T$, element $j$, and threshold $\tau$; furthermore, ADD is given access to the oracle $f$, the budget $k$, and the parameter $\delta$ of FIG. As an overview, ADD adds the first[3] element $x \geq j$, such that $x \notin T$ and such that the marginal gain $f_x(S)$ is at least $\tau$. If no such element $x \geq j$ exists, the threshold is decreased by a factor of $(1 - \delta)$ and the process is repeated (with $j$ set to 0). When such an element $x$ is found, the element $x$ is added to $S$, and the new threshold value and position $x$ are returned. Finally, ADD ensures that the size of $S$ does not exceed $k$.

Next, the approximation ratio of FIG is proven.

**Theorem 2.** *Let $f : 2^{[n]} \to \mathbb{R}^+$ be submodular, let $k \in [n]$, and let $\varepsilon > 0$. Let $O = \arg\max_{|S| \le k} f(S)$. Choose $\delta$ such that $(1 - 6\delta)/4 > 1/4 - \varepsilon$, and let $C = \mathtt{FIG}\,(f, k, \delta)$. Then*

$$f(C) \ge (1 - 6\delta)f(O)/4 \ge (1/4 - \varepsilon)\,f(O).$$

*Proof.* Let $A, B, C, D, E, M$ have their values at termination of $\mathtt{FIG}(f, k, \delta)$. Let $A = \{a_0, \dots, a_{|A|-1}\}$ be ordered by addition of elements by $\mathtt{FIG}$ into $A$. The proof requires the following four inequalities:

$$f(O \cup A) \le (2 + 2\delta)f(A) + \delta M, \tag{2}$$
$$f((O \setminus \{a_0\}) \cup B) \le (2 + 2\delta)f(B) + \delta M, \tag{3}$$
$$f(O \cup D) \le (2 + 2\delta)f(D) + \delta M, \tag{4}$$
$$f(O \cup E) \le (2 + 2\delta)f(E) + \delta M. \tag{5}$$

Once these inequalities have been established, Inequalities 2, 3, submodularity of $f$, and $A \cap B = \emptyset$ imply

$$f(O \setminus \{a_0\}) \le 2(1 + \delta)(f(A) + f(B)) + 2\delta M. \tag{6}$$

Similarly, from Inequalities 4, 5, submodularity of $f$, and $D \cap E = \{a_0\}$, it holds that

$$f(O \cup \{a_0\}) \le 2(1 + \delta)(f(D) + f(E)) + 2\delta M. \tag{7}$$

Hence, from the fact that either $a_0 \in O$ or $a_0 \notin O$ and the definition of $C$, it holds that

$$f(O) \le 4(1 + \delta)f(C) + 2\delta M.$$

Since $f(C) \le f(O)$ and $M \le f(O)$, the theorem is proved.

The proofs of Inequalities 2–5 are similar. The proof of Inequality 3 is given here, while the proofs of the others are provided in Appendix B.

*Proof of Inequality 3.* Let $A = \{a_0, \dots, a_{|A|-1}\}$ be ordered as specified by $\mathtt{FIG}$. Likewise, let $B = \{b_0, \dots, b_{|B|-1}\}$ be ordered as specified by $\mathtt{FIG}$.

**Lemma 1.** $O \setminus (B \cup \{a_0\}) = \{o_0, \dots, o_{l-1}\}$ *can be ordered such that*

$$f_{o_i}(B_i) \le (1 + 2\delta)f_{b_i}(B_i), \tag{8}$$

*for any $i \in [|B|]$.*

*Proof.* For each $i \in [|B|]$, define $\tau_{B_i}$ to be the value of $\tau$ when $b_i$ was added into $B$ by the $\mathtt{ADD}$ subroutine. Order $o \in (O \setminus (B \cup \{a_0\})) \cap A = \{o_0, \dots, o_{l-1}\}$ by the order in which these elements were added into $A$. Order the remaining elements of $O \setminus (B \cup \{a_0\})$ arbitrarily. Then, when $b_i$ w;as chosen by $\mathtt{ADD}$, it holds that $o_i \notin A_{i+1}$, since $A_1 = \{a_0\}$ and $a_0 \notin O \setminus (B \cup \{a_0\})$. Also, it holds that $o_i \notin B_i$ since $B_i \subseteq B$; hence $o_i$ was not added into some (possibly non-proper) subset $B_i'$ of $B_i$ at the previous threshold value $\frac{\tau_{B_i}}{(1-\delta)}$. By submodularity, $f_{o_i}(B_i) \le f_{o_i}(B_i') < \frac{\tau_{B_i}}{(1-\delta)}$. Since $f_{b_i}(B_i) \ge \tau_{B_i}$ and $\delta < 1/2$, inequality (8) follows.

Order $\hat{O} = O \setminus (B \cup \{a_0\}) = \{o_0, \dots, o_{l-1}\}$ as defined in the proof of Lemma 1, and let $\hat{O}_i = \{o_0, \dots, o_{i-1}\}$, if $i \ge 1$, and let $\hat{O}_0 = \emptyset$. Then

$$
\begin{aligned}
f(\hat{O} \cup B) - f(B) &= \sum_{i=0}^{l-1} f_{o_i}(\hat{O}_i \cup B) \\
&= \sum_{i=0}^{|B|-1} f_{o_i}(\hat{O}_i \cup B) + \sum_{i=|B|}^{l-1} f_{o_i}(\hat{O}_i \cup B) \\
&\le \sum_{i=0}^{|B|-1} f_{o_i}(B_i) + \sum_{i=|B|}^{l-1} f_{o_i}(B) \\
&\le \sum_{i=0}^{|B|-1} (1 + 2\delta)f_{b_i}(B_i) + \sum_{i=|B|}^{l-1} f_{o_i}(B) \\
&\le (1 + 2\delta)f(B) + \delta M,
\end{aligned}
$$

where any empty sum is defined to be 0; the first inequality follows by submodularity, the second follows from Lemma 1, and the third follows from the definition of $B$, and the facts that, for any $i$ such that $|B| \le i < l$, $\max_{x \in [n] \setminus A_{|B|+1}} f_x(B) < \delta M/k$, $l - |B| \le k$, and $o_i \notin A_{|B|+1}$.

**Theorem 3.** *Let $f : 2^{[n]} \to \mathbb{R}^+$ be submodular, let $k \in [n]$, and let $\delta > 0$. Then the number of queries to $f$ by $\mathtt{FIG}(f, k, \delta)$ is at most $O\left(\frac{n}{\delta} \log \frac{k}{\delta}\right)$.*

*Proof.* Recall $[n] = \{0, 1, \ldots, n - 1\}$. Let $S \in \{A, B, D, E\}$, and $S = \{s_0, \ldots, s_{|S|-1}\}$ in the order in which elements were added to $S$. When ADD is called by FIG to add an element $s_i \in [n]$ to $S$, if the value of $\tau$ is the same as the value when $s_{i-1}$ was added to $S$, then $s_i > s_{i-1}$. Finally, once ADD queries the marginal gain of adding $(n - 1)$, the threshold is revised downward by a factor of $(1 - \delta)$.

Therefore, there are at most $O(n)$ queries of $f$ at each distinct value of $\tau_A, \tau_B, \tau_D, \tau_E$. Since at most $O(\frac{1}{\delta} \log \frac{k}{\delta})$ values are assumed by each of these thresholds, the theorem follows. $\square$

## 3   Tight Examples

In this section, examples are provided showing that InterlaceGreedy or FastInterlaceGreedy may achieve performance ratio at most $1/4 + \varepsilon$ on specific instances, for each $\varepsilon > 0$. These examples show that the analysis in the preceding sections is tight.

Let $\varepsilon > 0$ and choose $k$ such that $1/k < \varepsilon$. Let $O$ and $D$ be disjoint sets each of $k$ distinct elements; and let $U = O \dot\cup \{a, b\} \dot\cup D$. A submodular function $f$ will be defined on subsets of $U$ as follows.

Let $C \subseteq U$.

- If both $a \in C$ and $b \in C$, then $f(C) = 0$.
- If $a \in C$ xor $b \in C$, then $f(C) = \frac{|C \cap O|}{2k} + \frac{1}{k}$.
- If $a \notin C$ and $b \notin C$, then $f(C) = \frac{|C \cap O|}{k}$.

The following proposition is proved in Appendix D.

**Proposition 1.** *The function $f$ is submodular.*

Next, observe that for any $o \in O$, $f_a(\emptyset) = f_b(\emptyset) = f_o(\emptyset) = 1/k$. Hence InterlaceGreedy or FastInterlaceGreedy may choose $a_0 = a$ and $b_0 = b$; after this choice, the only way to increase $f$ is by choosing elements of $O$. Hence $a_i, b_i$ will be chosen in $O$ until elements of $O$ are exhausted, which results in $k/2$ elements of $O$ added to each of $A$ and $B$. Thereafter, elements of $D$ will be chosen, which do not affect the function value. This yields

$$f(A) = f(B) \le 1/k + 1/4.$$

Next, $D_1 = E_1 = \{a\}$, and a similar situation arises, in which $k/2$ elements of $O$ are added to $D, E$, yielding $f(D) = f(E) = f(A)$. Hence InterlaceGreedy or FastInterlaceGreedy may return $A$, while $f(O) = 1$. So $\frac{f(A)}{f(O)} \le 1/k + 1/4 \le 1/4 + \varepsilon$.

## 4   Experimental Evaluation

In this section, performance of FastInterlaceGreedy (FIG) is compared with that of state-of-the-art algorithms on two applications of submodular maximization: cardinality-constrained maximum cut and network monitoring.

### 4.1   Setup

**Algorithms**  The following algorithms are compared.  Source code for the evaluated implementations of all algorithms is available at https://gitlab.com/kuhnle/non-monotone-max-cardinality.

- **FastInterlaceGreedy (Alg. 2)**: `FIG` is implemented as specified in the pseudocode, with the following addition: a stealing procedure is employed at the end, which uses submodularity to quickly steal[4] elements from $A, B, D, E$ into $C$ in $O(k)$ queries. This does not impact the performance guarantee, as the value of $C$ can only increase. The parameter $\delta$ is set to 0.1, yielding approximation ratio of 0.1.

- **Gupta et al. (2010)**: The algorithm of Gupta et al. (2010) for cardinality constraint; as the subroutine for the unconstrained maximization subproblems, the deterministic, linear-time $1/3$-approximation algorithm of Buchbinder et al. (2012) is employed. This yields an overall approximation ratio of $1/7$ for the implementation used herein. This algorithm is the fastest determistic approximation algorithm in prior literature.

- **FastRandomGreedy (FRG)**: The $O\left(\frac{n}{\varepsilon^2} \ln \frac{1}{\varepsilon}\right)$ randomized algorithm of Buchbinder et al. (2015) (Alg. 4 of that paper), with expected ratio $1/e - \varepsilon$; the parameter $\varepsilon$ was set to 0.3, yielding expected ratio of $\approx 0.07$ as evaluated herein. This algorithm is the fastest randomized approximation algorithm in prior literature.

- **BLITS**: The $O\left(\log^2 n\right)$-adaptive algorithm recently introduced in Balkanski et al. (2018); the algorithm is employed as a heuristic without performance ratio, with the same parameter choices as in Balkanski et al. (2018). In particular, $\varepsilon = 0.3$ and 30 samples are used to approximate the expectations. Also, a bound on OPT is guessed in logarithmically many iterations as described in Balkanski et al. (2018) and references therein.

Results for randomized algorithms are the mean of 10 trials, and the standard deviation is represented in plots by a shaded region.

**Applications**   Many applications with non-monotone, submodular objective functions exist. In this section, two applications are chosen to demonstrate the performance of the evaluated algorithms.

- Cardinality-Constrained Maximum Cut: The archetype of a submodular, non-monotone function is the maximum cut objective: given graph $G = (V, E)$, $S \subseteq V$, $f(S)$ is defined to be the number of edges crossing from $S$ to $V \setminus S$. The cardinality constrained version of this problem is considered in the evaluation.

- Social Network Monitoring: Given an online social network, suppose it is desired to choose $k$ users to monitor, such that the maximum amount of content is propagated through these users. Suppose the amount of content propagated between two users $u, v$ is encoded as weight $w(u, v)$. Then $f(S) = \sum_{u \in S, v \notin S} w(u, v)$.

## 4.2   Results

In this section, results are presented for the algorithms on the two applications. In overview: in terms of objective value, `FIG` and Gupta et al. (2010) were about the same and outperformed BLITS and FRG. Meanwhile, `FIG` was the fastest algorithm by the metric of queries to the objective and was faster than Gupta et al. (2010) by at least an order of magnitude.

**Cardinality Constrained MaxCut**   For these experiments, two random graph models were employed: an Erdős-Rényi (ER) random graph with $1,000$ nodes and edge probability $p = 1/2$, and a Barabási–Albert (BA) graph with $n = 10,000$ and $m = m_0 = 100$.

On the ER graph, results are shown in Figs. 1(a) and 1(b); the results on the BA graph are shown in Figs. 1(c) and 1(d). In terms of cut value, the algorithm of Gupta et al. (2010) performed the best, although the value produced by `FIG` was nearly the same. On the ER graph, the next best was FRG followed by BLITS; whereas on the BA graph, BLITS outperformed FRG in cut value. In terms of efficiency of queries, `FIG` used the smallest number on every evaluated instance, although the number did increase logarithmically with budget. The number of queries used by FRG was higher, but after a certain budget remained constant. The next most efficient was Gupta et al. (2010) followed by BLITS.

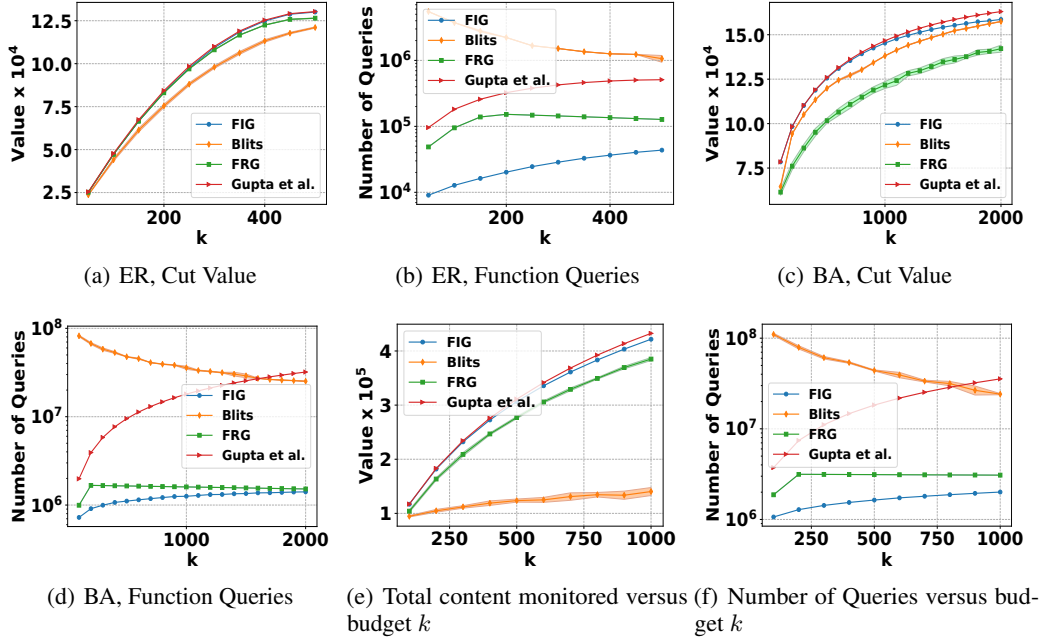

(a) ER, Cut Value

(b) ER, Function Queries

(c) BA, Cut Value

(d) BA, Function Queries

(e) Total content monitored versus budget $k$

(f) Number of Queries versus budget $k$

Figure 1: **(a)–(d)**: Objective value and runtime for cardinality-constrained maxcut on random graphs. **(e)–(f)**: Objective value and runtime for cardinality-constrained maxcut on ca-AstroPh with simulated amounts of content between users. In all plots, the $x$-axis shows the budget $k$.

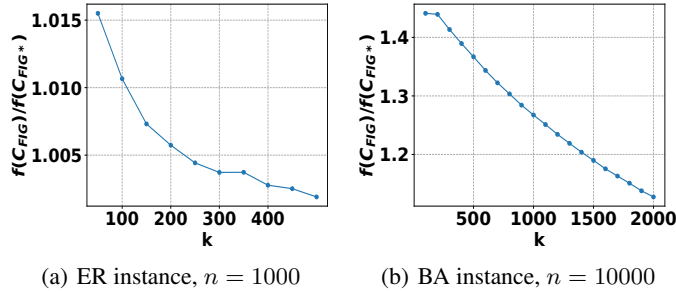

(a) ER instance, $n = 1000$

(b) BA instance, $n = 10000$

Figure 2: Effect of stealing procedure on solution quality of FIG.

**Social Network Monitoring**    For the social network monitoring application, the citation network ca-AstroPh from the SNAP dataset collection was used, with $n = 18,772$ users and $198,110$ edges. Edge weights, which represent the amount of content shared between users, were generated uniformly randomly in $[1, 10]$. The results were similar qualitatively to those for the unweighted MaxCut problem presented previously. FIG is the most efficient in terms of number of queries, and FIG is only outperformed in solution quality by Gupta et al. (2010), which required more than an order of magnitude more queries.

**Effect of Stealing Procedure**    In Fig. 2 above, the effect of removing the stealing procedure is shown on the random graph instances. Let $C_{FIG}$ be the solution returned by FIG, and $C_{FIG*}$ be the solution returned by FIG with the stealing procedure removed. Fig. 2(a) shows that on the ER instance, the stealing procedure adds at most $1.5\%$ to the solution value; however, on the BA instance, Fig. 2(b) shows that the stealing procedure contributes up to $45\%$ increase in solution value, although this effect degrades with larger $k$. This behavior may be explained by the interlaced greedy process being forced to leave good elements out of its solution, which are then recovered during the stealing procedure.

# 5    Acknowledgements

The work of A. Kuhnle was partially supported by Florida State University and the Informatics Institute of the University of Florida. Victoria G. Crawford and the anonymous reviewers provided helpful feedback which improved the paper.

## Footnotes

[1] The function $f$ is monotone if for all $A \subseteq B$, $f(A) \leq f(B)$.

[2]The mapping is to discard all elements greater than $n$.

[3]The first element $x > j$ in the natural ordering on $[n] = \{0, \ldots, n - 1\}$.

[4]Details of the stealing procedure are given in Appendix C.

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
