[Supplementary Material]

# Nearly Linear-Time, Deterministic Algorithm for Maximizing (Non-Monotone) Submodular Functions Under Cardinality Constraint

## Abstract

We develop two deterministic approximation algorithms for the maximization of non-monotone submodular functions under cardinality constraint: both are based upon the novel idea of interlacing two greedy procedures. Our algorithm FastInterlaceGreedy uses interlaced, thresholded greedy procedures to obtain

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

In this work, the problem studied is to maximize a submodular function under a cardinality constraint (MCC), which is formally defined as follows. Let $f : 2^n \to \mathbb{R}^+$ be submodular; let $k \in [n]$. Determine $A \subseteq [n]$ such that $|A| \leq k$ and for all $B$ such that $|B| \leq k$, $f(B) \leq f(A)$. An instance of MCC is the pair $(f, k)$; however, rather than an explicit description of $f$, the function $f$ is considered to be a value oracle; $f$ may be queried on any set $A \subseteq [n]$ to yield $f(A)$. The efficiency or runtime of an algorithm is measured by the number of queries made to the oracle $f$.

Finally, without loss of generality, instances of MCC considered in the following satisfy $n \geq 4k$. If this condition does not hold, the function may be extended to $[m]$ by adding dummy elements to the domain which do not change the function value. That is, the function $g : 2^m \to \mathbb{R}^+$ is defined as $g(A) = f(A \cap [n])$; it may be easily checked that $g$ remains submodular, and any possible solution to the MCC instance $(g, k)$ maps[2] to a solution of $(f, k)$ of the same value. Hence, the ratio of any solution to $(g, k)$ to the optimal is the same as the ratio of the mapped solution to the optimal on $(f, k)$.

## 2 Approximation Algorithms

In this section, we present the approximation algorithms based upon interlacing greedy procedures. In Section 2.1, the technique is demonstrated with standard greedy procedures in algorithm Interlace-Greedy. In Section 2.2, the nearly linear-time algorithm FastInterlaceGreedy is introduced.

### 2.1 The InterlaceGreedy Algorithm

In this section, the InterlaceGreedy algorithm (InterlaceGreedy, Alg. 1) is introduced. InterlaceGreedy takes as input an instance of MCC and outputs a set $C$, which approximates $\max_{|X| \leq k} f(X)$.

InterlaceGreedy operates by interlacing two standard ;j;greedy procedures. This interlacing is accomplished by maintaining two disjoint sets $A$ and $B$, which are initially empty. For $k$ iterations, the element $a \notin B$ with the highest marginal gain with respect to $A$ is added to $A$, followed by an analogous greedy selection for $B$; that is, the element $b \notin A$ with the highest marginal gain with respect to $B$ is added to $B$. After the first set of interlaced greedy procedures complete, a modified

**Algorithm 1** InterlaceGreedy $(f, k)$: The InterlaceGreedy Algorithm

---

1: **Input:** $f : 2^{[n]} \to \mathbb{R}^+$, $k \in [n]$
2: **Output:** $C \subseteq 2^{[n]}$, such that $|C| \le k$.
3: $A_0 \leftarrow B_0 \leftarrow \emptyset$
4: **for** $i \leftarrow 0$ to $k - 1$ **do**
5: $\quad a_i \leftarrow \arg\max_{x \in 2^{[n]} \setminus (A_i \cup B_i)} f_x(A_i)$
6: $\quad A_{i+1} \leftarrow A_i + a_i$
7: $\quad b_i \leftarrow \arg\max_{x \in 2^{[n]} \setminus (A_{i+1} \cup B_i)} f_x(B_i)$
8: $\quad B_{i+1} \leftarrow B_i + b_i$
9: $D_1 \leftarrow E_1 \leftarrow \{a_0\}$
10: **for** $i \leftarrow 1$ to $k - 1$ **do**
11: $\quad d_i \leftarrow \arg\max_{x \in 2^{[n]} \setminus (D_i \cup E_i)} f_x(D_i)$
12: $\quad D_{i+1} \leftarrow D_i + d_i$
13: $\quad e_i \leftarrow \arg\max_{x \in 2^{[n]} \setminus (D_{i+1} \cup E_i)} f_x(E_i)$
14: $\quad E_{i+1} \leftarrow E_i + e_i$
15: **return** $C \leftarrow \arg\max\{f(A_i), f(B_i), f(D_i), f(E_i) : i \in [k+1]\}$

---

version is repeated sets $D, E$, which are initialized to the maximum-value singleton $\{a_0\}$. Finally, the algorithm returns the set with the maximum $f$-value of any query the algorithm has made to $f$.

If $f$ is submodular, InterlaceGreedy has an approximation ratio of $1/4$ and query complexity $O(kn)$; the deterministic algorithm of Gupta et al. (2010) has the same time complexity to achieve ratio $1/6$. The full proof of Theorem 1 is provided in Appendix A.

**Theorem 1.** *Let $f : 2^{[n]} \to \mathbb{R}^+$ be submodular, let $k \in [n]$, let $O = \arg\max_{|S| \le k} f(S)$, and let $C =$InterlaceGreedy $(f, k)$. Then*

$$f(C) \ge f(O)/4,$$

*and InterlaceGreedy makes $O(kn)$ queries to $f$.*

*Proof sketch.* The main idea of the proof is to exploit the fact that if $S$ and $T$ are disjoint,

$$f(O \cup S) + f(O \cup T) \ge f(O) + f(O \cup S \cup T), \tag{1}$$

which is a consequence of the submodularity of $f$. Thus, if $f(S) \ge \alpha f(O \cup S)$ and $f(T) \ge \beta f(O \cup T)$, $\max_{X \in \{S, T\}} f(X) \ge (\alpha + \beta) f(O)/4$. Hence the proof proceeds by bounding $f(A) \ge f(O \cup A)/2$ and $f(B) \ge f((O \setminus \{a_0\}) \cup B)/2$. This is accomplished by an adaptation of the proof that the greedy algorithm is a $(1/2)$-approximation for monotone submodular maximization with respect to a matroid constraint (Fisher et al., 1978): the adaptation requires a re-ordering that is not possible subject to a general matroid constraint but is possible with the cardinality constraint considered here. Because of the way the re-ordering works, it is only possible to show that $f(B) \ge f((O \setminus \{a_0\}) \cup B)/2$, instead of the desired $f(B) \ge f(O \cup B)/2$. Hence, a second greedy interlacing is required, starting both sets from $\{a_0\}$, to produce $D, E$ such that $f(D) \ge f(O \cup D)/2$ and $f(E) \ge f(O \cup E)/2$, with $f(O \cup D) + f(O \cup E) \ge f(O \cup \{a_0\})$ by submodularity. Finally, the argument concludes by noticing that either $a_0 \in O$ or $a_0 \notin O$. $\qquad\square$

## 2.2 The FastInterlaceGreedy Algorithm

In this section, we provide a faster interlaced greedy algorithm (FastInterlaceGreedy (FIG), Alg. 2), which requires $O(n \log n)$ queries. As input, an instance $(f, k)$ of MCC is taken, as well as a parameter $\delta > 0$.

The algorithm FIG works as follows. As in InterlaceGreedy, there is a repeated interlacing of two greedy procedures. However, to ensure a faster query complexity, these greedy procedures are thresholded: a separate threshold $\tau$ is maintained for each of the greedy procedures. The interlacing is accomplished by alternating calls to the ADD subroutine (Alg. 3), which adds a single element and is described below. When all of the thresholds fall below the value $\delta M/n$, the maximum of the greedy solutions is returned; here, $\delta > 0$ is the input parameter, $M$ is the maximum value of a singleton, and $n$ is the size of the ground set.

---

**Algorithm 2** FIG $(f, k, \delta)$: The FastInterlaceGreedy Algorithm

---

1: **Input:** $f : 2^{[n]} \to \mathbb{R}^+$, $k \in [n]$
2: **Output:** $C \subseteq 2^{[n]}$, such that $|C| \leq k$.
3: $A_0 \leftarrow B_0 \leftarrow \emptyset$
4: $M \leftarrow \tau_A \leftarrow \tau_B \leftarrow \max_{x \in [n]} f(x)$
5: $i \leftarrow -1, a_{-1} \leftarrow 0, b_{-1} \leftarrow 0$
6: **while** $\tau_A \geq \varepsilon M/n$ or $\tau_B \geq \varepsilon M/n$ **do**
7:    $(a_{i+1}, \tau_A) \leftarrow$ ADD$(A, B, a_i, \tau_A)$
8:    $(b_{i+1}, \tau_B) \leftarrow$ ADD$(B, A, b_i, \tau_B)$
9:    $i \leftarrow i + 1$
10: $D_1 \leftarrow E_1 \leftarrow \{a_0\}, \tau_D \leftarrow \tau_E \leftarrow M$
11: $i \leftarrow 0, d_0 \leftarrow 0, e_0 \leftarrow 0$
12: **while** $\tau_D \geq \varepsilon M/n$ or $\tau_E \geq \varepsilon M/n$ **do**
13:    $(d_{i+1}, \tau_D) \leftarrow$ ADD$(D, E, d_i, \tau_D)$
14:    $(e_{i+1}, \tau_E) \leftarrow$ ADD$(E, D, e_i, \tau_E)$
15:    $i \leftarrow i + 1$
16: **return** $C \leftarrow \arg\max\{f(A), f(B), f(D), f(E)\}$

---

**Algorithm 3** ADD $(S, T, j, \tau)$: The ADD subroutine

---

1: **Input:** Two sets $S, T \subseteq [n]$, element $j \in [n]$, $\tau \in \mathbb{R}^+$
2: **Output:** $(i, \tau)$, such that $i \in [n]$, $\tau \in \mathbb{R}^+$
3: **if** $|S| = k$ **then**
4:    **return** $(0, (1 - \delta)\tau)$
5: **while** $\tau \geq \varepsilon M/n$ **do**
6:    **for** $(x \leftarrow j; x < n; x \leftarrow x + 1)$ **do**
7:      **if** $x \notin T$ **then**
8:        **if** $f_x(S) \geq \tau$ **then**
9:          $S \leftarrow S \cup \{x\}$
10:          **return** $(x, \tau)$
11:    $\tau \leftarrow (1 - \delta)\tau$
12:    $j \leftarrow 0$
13: **return** $(0, \tau)$

---

The ADD subroutine is responsible for adding a single element above the input threshold and decreasing the threshold. It takes as input four parameters: two sets $S, T$, element $j$, and threshold $\tau$; furthermore, ADD is given access to the oracle $f$, the budget $k$, and the parameter $\delta$ of FIG. As an overview, ADD adds the first[3] on the element $x > j$, such that $x \notin T$ and such that the marginal gain $f_x(S)$ is at least $\tau$. If no such element $x > j$ exists, the threshold is decreased by a factor of $(1 - \delta)$ and the process is repeated (with $j$ set to 0). When such an element $x$ is found, the element $x$ is added to $S$, and the new threshold value and position $x$ are returned. Finally, ADD ensures that the size of $S$ does not exceed $k$.

Next, we prove the approximation ratio of FIG.

**Theorem 2.** *Let $f : 2^{[n]} \to \mathbb{R}^+$ be submodular, let $k \in [n]$, and let $\varepsilon > 0$. Let $O = \arg\max_{|S| \leq k} f(S)$. Choose $\delta$ such that $(1 - 6\delta)/4 > 1/4 - \varepsilon$, and let $C =$FIG $(f, k, \delta)$. Then*

$$f(C) \geq (1 - 6\delta)f(O)/4 \geq (1/4 - \varepsilon) f(O).$$

*Proof.* Let $A, B, C, D, E, M$ have their values at termination of FIG$(f, k, \delta)$. Let $A = \{a_0, \ldots, a_{|A|-1}\}$ be ordered by addition of elements by FIG into $A$. The proof requires the following four inequalities:

$$f(O \cup A) \leq (2 + 2\delta)f(A) + \delta M, \tag{2}$$
$$f((O \setminus \{a_0\}) \cup B) \leq (2 + 2\delta)f(B) + \delta M, \tag{3}$$
$$f(O \cup D) \leq (2 + 2\delta)f(D) + \delta M, \tag{4}$$
$$f(O \cup E) \leq (2 + 2\delta)f(E) + \delta M. \tag{5}$$

Once these inequalities have been established, Inequalities 2, 3, submodularity of $f$, and $A \cap B = \emptyset$ imply

$$f(O \setminus \{a_0\}) \leq 2(1 + \delta)(f(A) + f(B)) + 2\delta M. \tag{6}$$

Similarly, from Inequalities 4, 5, submodularity of $f$, and $D \cap E = \{a_0\}$, it holds that

$$f(O \cup \{a_0\}) \leq 2(1 + \delta)(f(D) + f(E)) + 2\delta M. \tag{7}$$

Hence, from the fact that either $a_0 \in O$ or $a_0 \notin O$ and the definition of $C$, it holds that

$$f(O) \leq 4(1 + \delta)f(C) + 2\delta M.$$

Since $f(C) \leq f(O)$ and $M \leq f(O)$, the theorem is proved.

The proofs of Inequalities 2–5 are similar. The proof of Inequality 3 is given here, while the proofs of the others are provided in Appendix B.

*Proof of Inequality 3.* Let $A = \{a_0, \ldots, a_{|A|-1}\}$ be ordered as specified by `FIG`. Likewise, let $B = \{b_0, \ldots, b_{|B|-1}\}$ be ordered as specified by `FIG`.

**Lemma 1.** $O \setminus (B \cup \{a_0\}) = \{o_0, \ldots, o_{l-1}\}$ *can be ordered such that*

$$f_{o_i}(B_i) \leq (1 + 2\delta)f_{b_i}(B_i), \tag{8}$$

*for any $i \in [|B|]$.*

*Proof.* For each $i \in [|B|]$, define $\tau_{B_i}$ to be the value of $\tau$ when $b_i$ was added to $B$ in the ADD subroutine. Order $o \in (O \setminus (B \cup \{a_0\})) \cap A = \{o_0, \ldots, o_{\ell-1}\}$ by the order in which these elements were added into $A$. Order the remaining elements of $O \setminus (B \cup \{a_0\})$ arbitrarily. Then, when $b_i$ was chosen by ADD, it holds that $o_i \notin A_{i+1}$, since $A_1 = \{a_0\}$ and $a_0 \notin O \setminus (B \cup \{a_0\})$. Also, it is true $o_i \notin B_i$; hence $o_i$ was not added into some (possibly non-proper) subset $B_i'$ of $B_i$ at the previous threshold value $\frac{\tau_{B_i}}{(1-\delta)}$. Hence $f_{o_i}(B_i) \leq f_{o_i}(B_i') < \frac{\tau_{B_i}}{(1-\delta)}$, since $o_i \notin A_{i+1}$. Since $f_{b_i}(B_i) \geq \tau_{B_i}$ and $\delta < 1/2$, inequality (8) follows.

Order $\hat{O} = O \setminus (B \cup \{a_0\}) = \{o_0, \ldots, o_{l-1}\}$ as indicated in the proof of Lemma 1, and let $\hat{O}_i = \{o_0, \ldots, o_{i-1}\}$, if $i \geq 1$, $\hat{O}_0 = \emptyset$. Then

$$\begin{aligned}
f(\hat{O} \cup B) - f(B) &= \sum_{i=0}^{l-1} f_{o_i}(\hat{O}_i \cup B) \\
&= \sum_{i=0}^{|B|-1} f_{o_i}(\hat{O}_i \cup B) + \sum_{i=|B|}^{l-1} f_{o_i}(\hat{O}_i \cup B) \\
&\leq \sum_{i=0}^{|B|-1} f_{o_i}(B_i) + \sum_{i=|B|}^{l-1} f_{o_i}(B) \\
&\leq \sum_{i=0}^{|B|-1} (1 + 2\delta)f_{b_i}(B_i) + \sum_{i=|B|}^{l-1} f_{o_i}(B) \\
&\leq (1 + 2\delta)f(B) + \delta M,
\end{aligned}$$

where any empty sum is defined to be 0; the first inequality follows by submodularity, the second follows from Lemma 1, and the third follows from the definition of $B$, and the facts that $\max_{x \in [n] \setminus A_{|B|+1}} f_x(B) < \varepsilon M/n$, $l - |B| \leq k$, and $o_i \notin A_{|B|+1}$, for $|B| \leq i < l$.

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

## Footnotes

[1]For technical definitions of terms used in the Introduction, the reader is referred to Section 1.

[2]The mapping is to discard all elements greater than $n$.

[3]The first element $x > j$ in the natural ordering on $[n] = \{0, \ldots, n - 1\}$.

[4]Details of the stealing procedure are given in Appendix C.

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

## A  Proof of Theorem 1

*Proof of Theorem 1.*

**Lemma 2.**
$$4f(C) \geq f\left(O \setminus \{a_0\}\right).$$

*Proof.* Let $A = \arg \max_{i \in [k+1]} f(A_i)$. Let $\hat{O} = O \setminus A_k = \{o_0, \dots, o_{l-1}\}$ be ordered such that for each $i \in [l]$, $o_i \notin B_i$; this ordering is possible since $B_0 = \emptyset$ and $l \leq k$. Also, for each $i \in [l]$, let $\hat{O}_i = \{o_0, \dots, o_i\}$, and let $\hat{O}_0 = \emptyset$. Then

$$
\begin{aligned}
f(O \cup A_k) - f(A_k) &= \sum_{i=0}^{l-1} f_{o_i}(\hat{O}_i \cup A_k) \\
&\leq \sum_{i=0}^{l-1} f_{o_i}(A_i) \\
&\leq \sum_{i=0}^{l-1} f_{a_i}(A_i) = f(A_l),
\end{aligned}
$$

where the first inequality follows from submodularity, the second inequality follows from the greedy choice $a_i = \arg \max_{x \in 2^{[n]} \setminus (A_i \cup B_i)} f_x(A_i)$ and the fact that $o_i \notin B_i$. Hence

$$f(O \cup A_k) \leq f(A_l) + f(A_k) \leq 2f(A). \tag{9}$$

Let $B = \arg \max_{i \in [k+1]} f(B_i)$. Let $\hat{O} = O \setminus (\{a_0\} \cup B_k) = \{o_0, \dots, o_{l-1}\}$ be ordered such that for each $i \in [l]$, $o_i \notin A_{i+1}$; this ordering is possible since $A_1 = \{a_0\}$, $a_0 \notin \hat{O}$, and $l \leq k$. Also, for each $i \in [l]$, let $\hat{O}_i = \{o_0, \dots, o_i\}$, and let $\hat{O}_0 = \emptyset$. Then

$$
\begin{aligned}
f((O \setminus \{a_0\}) \cup B_k) - f(B_k) &= \sum_{i=0}^{l-1} f_{o_i}(\hat{O}_i \cup B_k) \\
&\leq \sum_{i=0}^{l-1} f_{o_i}(B_i) \\
&\leq \sum_{i=0}^{l-1} f_{b_i}(B_i) = f(B_l),
\end{aligned}
$$

where the first inequality follows from submodularity, the second inequality follows from the greedy choice $b_i = \arg \max_{x \in 2^{[n]} \setminus (A_{i+1} \cup B_i)} f_x(B_i)$ and the fact that $o_i \notin A_{i+1}$. Hence

$$f((O \setminus \{a_0\}) \cup B_k) \leq f(B_l) + f(B_k) \leq 2f(B). \tag{10}$$

By inequalities (9), (10), the fact that $A_k \cap B_k = \emptyset$, and submodularity, we have

$$f(O \setminus \{a_0\}) \leq f(O \cup A_k) + f((O \setminus \{a_0\} \cup B_k) \leq 2(f(A) + f(B)) \leq 4f(C).$$

$\square$

**Lemma 3.**
$$4f(C) \geq f\left(O \cup \{a_0\}\right).$$

*Proof.* Let $D = \arg \max_{i \in [k+1]} f(A_i)$. Let $\hat{O} = O \setminus D_k = \{o_0, \dots, o_{l-1}\}$ be ordered such that for each $i \in [l]$, $o_i \notin E_i$; this ordering is possible since $E_0 = \emptyset$ and $l \leq k$. Also, for each $i \in [l]$, let

$\hat{O}_i = \{o_0, \dots, o_i\}$, and let $\hat{O}_0 = \emptyset$. Then

$$
\begin{aligned}
f(O \cup D_k) - f(D_k) &= \sum_{i=0}^{l-1} f_{o_i}(\hat{O}_i \cup D_k) \\
&\leq \sum_{i=0}^{l-1} f_{o_i}(D_i) \\
&\leq \sum_{i=0}^{l-1} f_{d_i}(D_i) = f(D_l),
\end{aligned}
$$

where the first inequality follows from submodularity, the second inequality follows from the greedy choice $d_i = \arg\max_{x \in 2^{[n]} \setminus (D_i \cup E_i)} f_x(D_i)$ and the fact that $o_i \notin E_i$. Hence

$$f(O \cup D_k) \leq f(D_l) + f(D_k) \leq 2f(D). \tag{11}$$

Let $E = \arg\max_{i \in [k+1]} f(E_i)$. Let $\hat{O} = O \setminus E_k = \{o_0, \dots, o_{l-1}\}$ be ordered such that for each $i \in [l]$, $o_i \notin D_{i+1}$; this ordering is possible since $D_1 = \{a_0\}$, $a_0 \notin \hat{O}$ (since $a_0 \in E_k$), and $l \leq k$. Also, for each $i \in [l]$, let $\hat{O}_i = \{o_0, \dots, o_i\}$, and let $\hat{O}_0 = \emptyset$. Then

$$
\begin{aligned}
f(O \cup E_k) - f(E_k) &= \sum_{i=0}^{l-1} f_{o_i}(\hat{O}_i \cup E_k) \\
&\leq \sum_{i=0}^{l-1} f_{o_i}(E_i) \\
&\leq \sum_{i=0}^{l-1} f_{e_i}(E_i) = f(E_l),
\end{aligned}
$$

where the first inequality follows from submodularity, the second inequality follows from the greedy choices $e_0 = \arg\max_{x \in [n]} f(x)$, and if $i > 0$, $e_i = \arg\max_{x \in 2^{[n]} \setminus (D_{i+1} \cup E_i)} f_x(E_i)$ and the fact that $o_i \notin D_{i+1}$. Hence

$$f((O \cup E_k) \leq f(E_l) + f(E_k) \leq 2f(E). \tag{12}$$

By inequalities (11), (12), the fact that $D_k \cap E_k = \{a_0\}$, and submodularity, we have

$$f(O \cup \{a_0\}) \leq f(O \cup D_k) + f((O \cup E_k) \leq 2(f(D) + f(E)) \leq 4f(C).$$

$\square$

The proof of the theorem follows from Lemmas 2, 3, and the fact that one of the statements $a_0 \in O$ or $a_0 \notin O$ must hold; hence, either $O \cup \{a_0\} = O$ or $O \setminus \{a_0\} = O$. $\square$

# B  Proofs for Theorem 2

*Proof of Inequality 2.* Let $A = \{a_0, \dots, a_{|A|-1}\}$ be ordered as specified by FIG. Likewise, let $B = \{b_0, \dots, b_{|B|-1}\}$ be ordered as specified by FIG.

**Lemma 4.** $O \setminus A = \{o_0, \dots, o_{l-1}\}$ *can be ordered such that*

$$f_{o_i}(A_i) \leq (1 + 2\delta) f_{a_i}(A_i), \tag{13}$$

*if $i \in [|A|]$.*

*Proof.* Order $o \in (O \setminus A) \cap B = \{o_0, \dots, o_{\ell-1}\}$ by the order in which these elements were added into $B$. Order the remaining elements of $O \setminus A$ arbitrarily. Then, when $a_i$ was chosen by ADD, it holds that $o_i \notin B_i$. Also, it is true $o_i \notin A_i$; hence $o_i$ was not added into some (possibly non-proper) subset $A_i'$ of $A_i$ at the previous threshold value $\frac{\tau_{A_i}}{(1-\delta)}$. Hence $f_{o_i}(A_i) \leq f_{o_i}(A_i') < \frac{\tau_{A_i}}{(1-\delta)}$, since $o_i \notin B_i$. Since $f_{a_i}(A_i) \geq \tau_{A_i}$ and $\delta < 1/2$, inequality (13) follows. $\square$

Order $\hat{O} = O \setminus A = \{o_0, \dots, o_{l-1}\}$ as indicated in the proof of Lemma 4, and let $\hat{O}_i = \{o_0, \dots, o_{i-1}\}$, if $i \geq 1$, $\hat{O}_0 = \emptyset$. Then

$$
\begin{aligned}
f(O \cup A) - f(A) &= \sum_{i=0}^{l-1} f_{o_i}(\hat{O}_i \cup A) \\
&= \sum_{i=0}^{|A|-1} f_{o_i}(\hat{O}_i \cup A) + \sum_{i=|A|}^{l-1} f_{o_i}(\hat{O}_i \cup A) \\
&\leq \sum_{i=0}^{|A|-1} f_{o_i}(A_i) + \sum_{i=|A|}^{l-1} f_{o_i}(A) \\
&\leq \sum_{i=0}^{|A|-1} (1+2\delta) f_{a_i}(A_i) + \sum_{i=|A|}^{l-1} f_{o_i}(A) \\
&\leq (1+2\delta) f(A) + \delta M,
\end{aligned}
$$

where any empty sum is defined to be 0; the first inequality follows by submodularity, the second follows from Lemma 4, and the third follows from the definition of $A$, and the facts that $\max_{x \in [n] \setminus B_{|A|}} f_x(A) < \varepsilon M/n$ and $l - |A| \leq k$. $\qquad\square$

*Proof of Inequality 4.* As in the proof of Inequality 2, it suffices to establish the following lemma. $\qquad\square$

**Lemma 5.** $O \setminus D = \{o_0, \dots, o_{l-1}\}$ *can be ordered such that*

$$
f_{o_i}(D_i) \leq (1+2\delta) f_{d_i}(D_i), \tag{14}
$$

*for $i \in [|D|]$.*

*Proof.* Order $o \in (O \setminus D) \cap E = \{o_0, \dots, o_{\ell-1}\}$ by the order in which these elements were added into $E$. Order the remaining elements of $O \setminus D$ arbitrarily. Then, when $d_i$ was chosen by ADD, it holds that $o_i \notin E_i$. Also, it is true $o_i \notin D_i$; hence $o_i$ was not added into some (possibly non-proper) subset $D_i'$ of $D_i$ at the previous threshold value $\frac{\tau_{D_i}}{(1-\delta)}$. Hence $f_{o_i}(D_i) \leq f_{o_i}(D_i') < \frac{\tau_{D_i}}{(1-\delta)}$, since $o_i \notin E_i$. Since $f_{d_i}(D_i) \geq \tau_{D_i}$ and $\delta < 1/2$, inequality (14) follows. $\qquad\square$

*Proof of Inequality 5.* As in the proof of Inequality 2, it suffices to establish the following lemma.

**Lemma 6.** $O \setminus E = \{o_0, \dots, o_{l-1}\}$ *can be ordered such that*

$$
f_{o_i}(E_i) \leq (1+2\delta) f_{e_i}(E_i), \tag{15}
$$

*for $i \in [|E|]$.*

*Proof.* Order $o \in (O \setminus E) \cap D = \{o_0, \dots, o_{\ell-1}\}$ by the order in which these elements were added into $D$. Order the remaining elements of $O \setminus E$ arbitrarily. Then, when $e_i$ was chosen by ADD, it holds that $o_i \notin D_{i+1}$, since $D_1 = \{a_0\}$ and $a_0 = d_0 \notin O \setminus E$. Also, it is true $o_i \notin E_i$; hence $o_i$ was not added into some (possibly non-proper) subset $E_i'$ of $E_i$ at the previous threshold value $\frac{\tau_{E_i}}{(1-\delta)}$. Hence $f_{o_i}(E_i) \leq f_{o_i}(E_i') < \frac{\tau_{E_i}}{(1-\delta)}$, since $o_i \notin D_{i+1}$. Since $f_{e_i}(E_i) \geq \tau_{E_i}$ and $\delta < 1/2$, inequality (15) follows. $\qquad\square$

$\qquad\square$

# C   Stealing Procedure for FastInterlaceGreedy

In this section, we describe an $O(k)$ procedure that may improve the quality of the solution found by FastInterlaceGreedy (a similar procedure could also be employed for InterlaceGreedy).

Let $A, B, C, D, E$ have their values at the termination of FastInterlaceGreedy. Then calculate the sets $G = \{B_c = f(C) - f(C \setminus \{c\}) : c \in C\}$ and $H = \{A_x = f(C \cup \{x\}) - f(C) : x \in A \cup B \cup D \cup E\}$.

384 Then sort $G = (B_{c_1}, \ldots, B_{c_k})$ in non-decreasing order and sort $H = (A_{x_1}, \ldots, A_{x_l})$ in non-
385 increasing order. Computing and sorting these sets requires $O(k \log k)$ time (and only $O(k)$ queries
386 to $f$).

387 Finally, iterate through the elements of $G$ in the sorted order, and if $B_{c_i} < A_{x_i}$ then $C$ is assigned
388 $C \setminus \{c_i\} \cup \{x_i\}$ if this assignment increases the value $f(C)$.