[Reviews · NeurIPS 2019]

Reviewer 1



In this paper, the authors study the problem of maximizing a non-monotone submodular function subject to a cardinality constraint and present a deterministic algorithm that achieves (1/4 - \epsilon)-approximation for the problem. Their fastest algorithm makes O(n / \epsilon \log(n / \epsilon)) queries to the oracle. While Buchbinder et al. (2015) have designed a randomized algorithm with better approximation factor and time complexity, the algorithm presented in this paper is deterministic. Although I believe deterministic algorithms which guarantee the performance in the worst-case scenarios are interesting for many researchers in the field, the contribution level of this paper is borderline. Also, I checked almost all the proofs in this paper. As far as I have checked they are correct and rigorous. Next, I will provide a few comments which can improve the quality of this paper: Related work: There are several related works which are missing from this manuscript: 1. Maximizing non-monotone submodular functions subject to a matroid independence constraint (generalizes a cardinality constraint): [V2013] Jan Vondrák: Symmetry and Approximability of Submodular Maximization Problems. SIAM J. Comput. 42(1): 265-304 (2013). 2. Streaming algorithms for non-monotone submodular functions: [FKK2019] Moran Feldman, Amin Karbasi, Ehsan Kazemi: Do Less, Get More: Streaming Submodular Maximization with Subsampling. NeurIPS 2018: 730-740. 3. Improve the adaptive complexity for non-monotone submodular functions: [FMZ2019] Matthew Fahrbach, Vahab S. Mirrokni, Morteza Zadimoghaddam: Non-monotone Submodular Maximization with Nearly Optimal Adaptivity and Query Complexity. ICML 2019: 1833-1842 Experimental results: 1. The low adaptivity algorithm of [FMZ2019] could be beneficial to be compared with the algorithms presented in this paper. 2. The algorithms of Feldman et al. (2017) and [FKK2019] are designed for maximizing non-monotone submodular functions subject to more general constraints. It would be interesting to compare the FIG algorithm with them. 3. The difference between Social Network Monitoring function and Maximum Cut function is not clear to me. Code and reproducibility: While the authors have provided a link for the source code, I could not find any file at the following address: https://gofile.io/?c=ChYSOQ Minor comments: 1. Line 36: achieve 2. Line 119: ";j;greedy" 3. Lines 5 and 7 of Algorithm 1: x \in [n] 4. Line 321 proof of lemma 2: in the definition of \hat{O}_i the last element in the set should be o_{i-1}. This happens again at line 326. Please check the proof for these typos one more time. 5. The proof sketch after Theorem 1 is vague and difficult to follow. -------------------------------------------------- Thanks for the answers. I would really appreciate if you could improve the sketch of the proofs in the camera-ready version.

Reviewer 2



Quality The algorithms and corresponding analysis are nontrivial but straightforward technically. Proofs in the main text and appendix appear to be correct except for one detail: In the proof of Lemma 1, how is the last term in the second to last equation upper bounded by \delta M? This requires justification since the setup of Theorem 2 implies \epsilon > 2/3 \delta. If this is an error, I suspect the proof can be corrected without significantly affecting the main results. Question: For the Blitz algorithm, why do the number of queries to f consistently decrease for larger k? How much of the performance of FIG is due to the stealing trick (Appendix C)? Is the proposed algorithm still competitive with Gupta et al. without the stealing trick? Originality Interlacing idea appears to be novel, but the proof techniques build on previous work. For example reordering trick which is crucial for lifting the monotonicity assumption goes back to Nemhauser et al 1978 Clarity: Overall very clear, well organized, and a pleasure to read. One type on line 119, ";j;greedy" Significance: This is an important open theoretical problem, and the reordering trick may be applicable to other recent algorithms for monotone submodular maximization. The experimental results would be more significant with an application besides than cardinality-constrained (weighted) max-cut. Question: How does the algorithm perform on monotone submodular functions? ---------------- EDIT: The authors addressed my concerns about ablation on stealing trick and a small typo (Algorithm 2 and line 190). Therefore I am keeping my review at accept

Reviewer 3



As discussed above, I believe that the interlacing greedy method is the most novel contribution and the thresholding technique is also a very nice addition which allows for the nearly linear run time. I believe related work is adequately cited. I have verified the proofs and believe them to be technically sound. The submission is very well written and the prose is clear and easy to read. EDIT: The authors have addressed all of my concerns and I keep my score of accept. I believe the significance of this work is the simplicity and speed of the algorithms presented. Of course, this paper does not give improved approximation ratios for these problems (and indeed, better approximation ratios are possible), but these algorithms are more practical than some previously proposed in the literature. Below are some comments regarding the algorithms and experiments: Algorithms 1. Are there instances where the interlacing greedy or the fast variant achieve an approximation ratio of exactly 1/4? In other words, is 1/4 the best approximation to hope for with this technique? 2. The proofs of the algorithms are terse in a few places. In particular, I think the proof of theorem 2 would benefit from a little more discussion on the inequalities required at the end for dealing with the delta term. For instance, I worked out the inequality $(1-2 \delta)/(1 + \delta) >= 1 - 6 \delta$ for $\delta \in [0,1]$ which I think is being used. Although this is easy to verify, I think it’s worth some acknowledgement. Additionally, the authors may want to highlight where they use non-negativity of f in the proofs (although non-negativity is assumed in their definition of submodularity, it is still helpful to see). Finally, the use of submodular inequality f(X \cap Y) + f(X \cup Y) <= f(X) + f(Y) is used in the proof but not discussed in the definitions; it would be nice to include a brief discussion of this equivalent definition of submoularity so that unfamiliar readers may better follow the proof. Experiments: 1. Unfortunately, the link does not contain any code so I could not look at the code. I assume that the author(s) will fix this and it doesn’t greatly affect my review. 2. Can you comment on why might the run time be slowly increasing with k? It’s worth noting that this empirical performance does not contradict your run time analysis. For instance, the # evaluations may be increasing as a function of k, but bounded above by the O(n log n). 3. My understanding is that in this situation, it is incorrect to say that the query complexity is increasing logarithmically with k. Although we see that, empirically, the number of evaluations grows slowly with k, there has been no test to determine that growth is logarithmic. In fact, using the term “logarithmic” implies some precise statement which is probably not true. So, I recommend replacing the discussion about “logarithmic growth” with a discussion about the point above. 4. As per the NeurIPS 2019 reproducability checklist, it would be good to include the language used and computing infrastructure used, even though the metrics of interest (e.g. # function evaluations) are machine independent. General Typos / Suggestions 1. Line 23: Add the epsilon dependence to the work of Badanidiyuru and Vondrak. 2. Lines 100-102: It might be best to describe the problem by the notation $max_{|S| \leq k} f(S)$ rather than the way it is phrased here. 3. Line 102: “the function $f$ is considered to be a value oracle” is awkwardly phrased. The function itself is not a value oracle, but it is “presented as” or “accessed by” a value oracle. 4. Line 119: Type “two standard ;j; greedy procedures” 5. The use of [n] seems to be inconsistent. Sometimes, it refers to the integers 1…n but other times it seems to refer to the integers 0…n-1. For example, Line 1 and Line 15 of Algorithm 1, where [n] = 1…n and [k+1] = 0…k are both used. 6. Line 146: Include delta / epsilon dependence in the running time 7. Line 158: “adds the first on the element” awkward phrasing, might be a typo. 8. Section 2.2 uses epsilon and delta interchangeably. I would recommend using only one for clarity. EDIT: The authors have sufficiently addressed all of my concerns and also (in my opinion) the concerns of other reviewers. I maintain my vote of "accept".

[Author Response · NeurIPS 2019]



(a) ER instance, $n = 1000$          (b) BA instance, $n = 10000$

Figure 1: Ablation study, effect of stealing behavior.

We thank all reviewers for the comprehensive feedback. In the following, we address specific concerns of each reviewer
individually. In addition to the below items, the source code link and other minor issues pointed out by the reviewers
will be fixed in the next version.

**Reviewer 1**: There are several related works which are missing from this manuscript: . . . Thank you for bringing these
works to our attention. We will add discussion of these works to the manuscript.

[The algorithms of] [FMZ2019], Feldman et al. (2017), [FKK2019] could be beneficial to be compared with the
algorithms presented in this paper. We agree that more empirical comparisons, especially with [FMZ2019], would be
interesting. If we had been aware of this work earlier, it may have provided a better contrast than BLITS, since it has
low query complexity as well as adaptivity.

**Reviewer 2**: In the proof of Lemma 1, how is the last term in the second to last equation upper bounded by $\delta M$? All
usage of $\varepsilon$ in the pseudocode of FIG, ADD should be $\delta$, which then yields the upper bound. Thank you for catching this
typo.

For the BLITS algorithm, why do the number of queries to f consistently decrease for larger k? BLITS works by
guessing successively smaller values for OPT. With larger $k$, it is able to terminate earlier as the value of any solution it
obtains is a lower bound on OPT.

How much of the performance of FIG is due to the stealing trick (Appendix C)? Is the proposed algorithm still
competitive with Gupta et al. without the stealing trick? In Fig. 1 above, we show the effect of removing the stealing
procedure on the random graph instances evaluated in the manuscript. Let $C_{FIG}$ be the solution returned by FIG, and
$C_{FIG*}$ be the solution returned by FIG with the stealing procedure removed. Fig. 1(a) shows that on the ER instance,
the stealing procedure adds at most $1.5\%$ to the solution value; however, on the BA instance, Fig. 1(b) shows that the
stealing procedure contributes up to $45\%$ increase in solution value, although this effect degrades with larger $k$. This
behavior may be explained by the interlaced greedy process being forced to leave good elements out of its solution,
which are then recovered during the stealing procedure.

How does the algorithm perform on monotone submodular functions? If the submodular function is restricted to be
monotone, IG obtains an approximation ratio of at least $1/2$. A factor of 2 is saved from the analysis for general
submodular, since $f(O \cup A) + f(O \cup B) \geq 2f(O)$ in the monotone case. It is possible a better analysis could show a
ratio of better than $1/2$. We did not evaluate empirically on monotone submodular functions.

**Reviewer 3**: Are there instances where the interlacing greedy or the fast variant achieve an approximation ratio of
exactly 1/4? In other words, is 1/4 the best approximation to hope for with this technique? Yes, the ratio $1/4$ is tight
for non-monotone submodular functions. Tight examples will be added to the manuscript showing, for each $\epsilon > 0$, an
instance $A_\epsilon$ where InterlaceGreedy has ratio less than $1/4 + \epsilon$ on $A_\epsilon$.

The proofs of the algorithms are terse in a few places. . . . We agree, and we will add the details indicated, the equivalent
definition of submodularity, and highlight the places where non-negativity is used.

Can you comment on why might the run time be slowly increasing with k? In FIG, we terminate a greedy subroutine
when the marginal gain falls below $\delta M/n$, which leads to the $\log n$ term in the query complexity. However, it is
sufficient in the proof to terminate when the gain is below $\delta M/k$, which would lead to a $\log k$ term in the query
complexity. We will update the algorithm and discussion to reflect this fact.

[Meta-Review · NeurIPS 2019]

This is a nice contribution on deterministic (non-monotone) submodular maximization subject to a size constraint. All the reviewers liked the paper. Please add the missing references to the final version.